Scientific Report

EMBO
reports

# Centromere localization and function of Mis18 requires Yippee-like domain-mediated oligomerization

Lakxmi Subramanian[†], Bethan Medina-Pritchard[†], Rachael Barton, Frances Spiller, Raghavendran Kulasegaran-Shylini, Guoda Radaviciute, Robin C Allshire & A Arockia Jeyaprakash[*]

## Abstract

Mis18 is a key regulator responsible for the centromere localization of the CENP-A chaperone Scm3 in *Schizosaccharomyces pombe* and HJURP in humans, which establishes CENP-A chromatin that defines centromeres. The molecular and structural determinants of Mis18 centromere targeting remain elusive. Here, by combining structural, biochemical, and yeast genetic studies, we show that the oligomerization of *S. pombe* Mis18, mediated via its conserved N-terminal Yippee-like domain, is crucial for its centromere localization and function. The crystal structure of the N-terminal Yippee-like domain reveals a fold containing a cradle-shaped pocket that is implicated in protein/nucleic acid binding, which we show is required for Mis18 function. While the N-terminal Yippee-like domain forms a homodimer *in vitro* and *in vivo*, full-length Mis18, including the C-terminal α-helical domain, forms a homotetramer *in vitro*. We also show that the Yippee-like domains of human Mis18α/Mis18β interact to form a heterodimer, implying a conserved structural theme for Mis18 regulation.

**Keywords** CENP-A; centromere; epigenetics; Mis18; Yippee
**Subject Categories** Cell Cycle; DNA Replication, Repair & Recombination; Structural Biology

## Introduction

The accurate distribution of genetic information to daughter cells during cell division relies on the physical attachment of chromosomes to spindle microtubules mediated by kinetochores. Kinetochores are large protein assemblies deposited at specific chromosomal loci known as centromeres [1–3]. Defective centromere function results in chromosome segregation errors that can contribute to genomic instability implicated in cancer [4]. Hence, understanding the molecular mechanisms that promote kinetochore establishment and maintenance at centromeres is of prime importance.

The location of most eukaryotic centromeres is determined by the assembly of specialized chromatin composed of nucleosomes in which canonical histone H3 is replaced by the centromere-specific H3 variant CENP-A in vertebrates and Cnp1 (CENP-A$^{\text{Cnp1}}$) in *Schizosaccharomyces pombe* [3,5]. Thus, the establishment and maintenance of kinetochores requires CENP-A to be recruited to and deposited at centromeres. In *S. pombe*, CENP-A$^{\text{Cnp1}}$ is specifically incorporated into chromatin over the central domain of endogenous centromeres where it is flanked by heterochromatin formed on outer repeat elements [6,7].

During S phase, CENP-A$^{\text{Cnp1}}$ levels at fission yeast centromeres are halved following DNA replication. Subsequently, CENP-A$^{\text{Cnp1}}$ levels are replenished during G2 phase of the cell cycle [8]. In early mitosis, the Mis18 complex comprising Mis18, Mis16, Eic1/Mis19/Kis1, and Eic2/Mis20, along with the CENP-A$^{\text{Cnp1}}$ chaperone Scm3 (counterpart of vertebrate HJURP; HJURP$^{\text{Scm3}}$), dissociates from centromeres and re-associates in mid/late anaphase following chromosome segregation [9–15]. Exclusion of the Mis18 complex and HJURP$^{\text{Scm3}}$ from centromeres during mitosis likely provides an opportunity for the CENP-A$^{\text{Cnp1}}$ loading cycle to reset and thereby prevent continual CENP-A$^{\text{Cnp1}}$ deposition [16]. HJURP$^{\text{Scm3}}$ directly interacts with CENP-A$^{\text{Cnp1}}$ and is essential for the deposition of new CENP-A$^{\text{Cnp1}}$ at centromeres [14]. Genetic studies have shown that Mis16, Mis18, and Eic1/Mis19/Kis1 are essential genes that are required for the localization of HJURP$^{\text{Scm3}}$ to centromeres and hence CENP-A$^{\text{Cnp1}}$ maintenance at centromeres [10,11,13,14,17].

Most of the components of the CENP-A assembly pathway are conserved among eukaryotes, but with a few striking differences. Humans possess two isoforms of Mis18 (Mis18α and Mis18β) and Mis16 (RbAp46 and RbAp48) [10]. During telophase, the human Mis18 complex comprising Mis18α, Mis18β, and Mis18BP1/KNL2, along with RbAp46 and RbAp48, associates with centromeres [17]. Mis18BP1/KNL2 has no detectable *S. pombe* homolog, but Eic1/Mis19/Kis1 appears to perform an analogous function [11,13]. As in

Wellcome Trust Centre for Cell Biology, Institute of Cell Biology, University of Edinburgh, Edinburgh, UK
*Corresponding author. Tel: +44 131 6507113; E-mail: jeyaprakash.arulanandam@ed.ac.uk
†These authors contributed equally to this work

*S. pombe*, the human Mis18 complex is required for HJURP recruitment to centromeres, where CENP-A is deposited during early G1 rather than G2 [9]. In addition, the stable incorporation of CENP-A at human centromeres requires the small GTPase activity of Cdc42 regulated by MgcRacGap/ECT2 [18].

Although key conserved players involved in the assembly and maintenance of CENP-A chromatin at centromeres have been identified, the molecular mechanisms through which they exert their function remain unclear. Mis18 is critical for the specification of centromeres from fission yeast to humans, however, what allows Mis18 to regulate centromere specification remains largely unknown. To gain insights into the structural features of *S. pombe* Mis18 that allow it to bind centromeres and promote HJURP[Scm3] recruitment and CENP-A[Cnp1] assembly, we determined the crystal structure of its highly conserved "Yippee-like" N-terminal globular domain. Our structural and biochemical analyses reveal that the Mis18 "Yippee-like" domain possesses a fold that is implicated in protein/nucleic acid binding and that this domain has an innate tendency to homodimerize both *in vitro* and *in vivo*. However, full-length Mis18 forms a homotetramer *in vitro,* highlighting a role for the C-terminal α-helical domain in influencing the overall oligomeric state of the protein. Genetic analyses using structure-guided mutants demonstrate that dimerization of the "Yippee-like" domain is essential for the centromere localization and hence the function of Mis18.

## Results and Discussion

### *sp*Mis18 possesses an N-terminal Yippee-like globular domain that is implicated in protein/nucleic acid binding

Amino acid sequence and the predicted secondary structure analysis of *Schizosaccharomyces pombe* Mis18 suggested the presence of a highly conserved N-terminal globular domain (residues 1–120; $sp\text{Mis18}_{1–120}$) mainly comprised of β-strands, followed by a moderately conserved C-terminal α-helical domain (residues 121–end; $sp\text{Mis18}_{\text{C-term-}\alpha}$) (Fig 1A and B). $\text{Mis18}_{1–120}$ shares about 20% sequence similarity with a putative $Zn^{2+}$-binding protein of unknown function named Yippee, originally identified in *Drosophila*, that is well conserved from yeast to humans [19]. $sp\text{Mis18}_{1–120}$ has two conserved C-X-X-C motifs, which are signature motifs present in metal ion-binding proteins. Previous analysis showed that mutations within the C-X-X-C motifs of human Mis18α

perturb its centromere localization, highlighting the essential role of this domain [17]. To structurally characterize $sp\text{Mis18}_{1–120}$, we purified recombinant $sp\text{Mis18}_{1–120}$ as a mono-disperse sample and obtained crystals that diffracted X-rays to about 2.6 Å (Fig 1C). X-ray fluorescence scans of the crystals revealed the presence of bound $Zn^{2+}$ ions. The structure was determined by single anomalous dispersion (SAD) exploiting the Zn anomalous signal from the bound $Zn^{2+}$ ions in the crystals. The refined structure has an R factor of 22 and R-free factor of 26 with good stereochemistry (Table 1 and Fig EV1A). The final model includes amino acid residues 19–118 of $sp\text{Mis18}_{1–120}$ (Fig 1D). The N-terminal 18 and C-terminal 2 (119 and 120) residues are presumably disordered and hence could not be modeled.

The overall fold of $sp\text{Mis18}_{1–120}$ is formed by antiparallel β-sheets: a three-stranded (β1-β2-β9: β-sheet I) and a six-stranded (β3-β4-β8-β7-β6-β5: β-sheet II) sheet, arranged approximately perpendicular to each other (Fig 1D). The two β-sheets are held together by a $Zn^{2+}$ ion coordinated via the C-X-X-C motifs from loops L1 and L5 (Fig 1D). Structural comparison of $sp\text{Mis18}_{1–120}$ with the available structures in the protein data bank (PDB) identified the thalidomide-binding domain of Cereblon (PDB: 4tzc; Q-score [20]: 0.47; RMSD: 2.12 Å), a component of an E3 ubiquitin ligase complex implicated in DNA repair, replication, and transcription [21], as the closest structural homolog (Fig EV1B). Other proteins that share a similar fold include RIG-I (PDB: 2qfd; Q-score: 0.38; RMSD: 2.19 Å), a nucleic acid-binding protein involved in innate anti-viral immunity [22]; Mss4 (PDB: 1fwq; Q-score: 0.34; RMSD: 2.03 Å), a guanine nucleotide exchange factor [23]; and MsrB (methionine sulfoxide reductase-B, PDB: 3e0o; Q-score: 0.32; RMSD: 1.81 Å), an oxidative reductase implicated in aging [24] (Fig EV1B). This is in agreement with a recent bioinformatics study suggesting an evolutionary relationship between Cereblon, Yippee, and Mis18 proteins [25]. Although these proteins recognize substrates as diverse as nucleic acids to proteins, they do so via a common cradle-shaped binding pocket formed by β-sheet II (Fig EV1C and D). This observation suggested that the putative substrate-binding site of $sp\text{Mis18}_{1–120}$ might play an important role in Mis18 function. To test whether the putative substrate-binding pocket was required for $sp\text{Mis18}$ function *in vivo*, we tested the ability of additional $sp\text{Mis18}$ expressed from a plasmid to complement the growth phenotype of *mis18-262* (G117D) cells, which exhibit loss of function for $sp\text{Mis18}$ at the restrictive temperature (36°C) [10]. While expression of wild-type $sp\text{Mis18}$ restored growth at 36°C, expressing the pocket mutant (Y74A/Y90A/T105A/S107K,

**Figure 1.** ***sp*Mis18 possesses a highly conserved N-terminal Yippee-like globular domain and a C-terminal α-helical domain.**

- A  Schematic representation of domain organization of Mis18 proteins from *Schizosaccharomyces pombe* and human (as suggested by the Conserved Domain Database (CDD) and secondary structure predictions).
- B  Amino acid conservation of Mis18 proteins among eukaryotes. The alignment includes orthologs from *S. pombe* (*sp*), *G. gallus* (*gg*), *M. musculus* (*mm*), *H. sapiens* (*hs*), and *B. taurus* (*bt*). Predicted (light orange; using PsiPred, http://bioinf.cs.ucl.ac.uk/psipred) and observed (bright orange; from the crystal structure shown in D) secondary structure elements are shown below the aligned sequences. Amino acid residues mutated in this study are highlighted with circles (dimer interface residues) and asterisks (putative substrate-binding pocket residues).
- C  SDS–PAGE showing a representative fraction of the purified *sp*Mis18 N-terminal Yippee-like globular domain (amino acid residues 1–120; $sp\text{Mis18}_{1–120}$).
- D  Cartoon representation of the crystal structure of $sp\text{Mis18}_{1–120}$ in two different orientations, highlighting key residues within the putative substrate-binding pocket.
- E  Substrate-binding pocket mutations Y90A and Y74A/Y90A/T105A/S107K affect the ability of ectopically expressed $sp\text{Mis18}_{fl}$ to rescue the temperature sensitivity of *mis18-262* cells to varying degrees. Fivefold serial dilutions of *mis18-262* cells transformed with plasmids harboring the indicated $sp\text{Mis18}_{fl}$ constructs, spotted on PMG — uracil + phloxine B media supplemented with (repressed) or without (expressed) thiamine, and incubated at the indicated temperatures; dead cells stain dark pink.

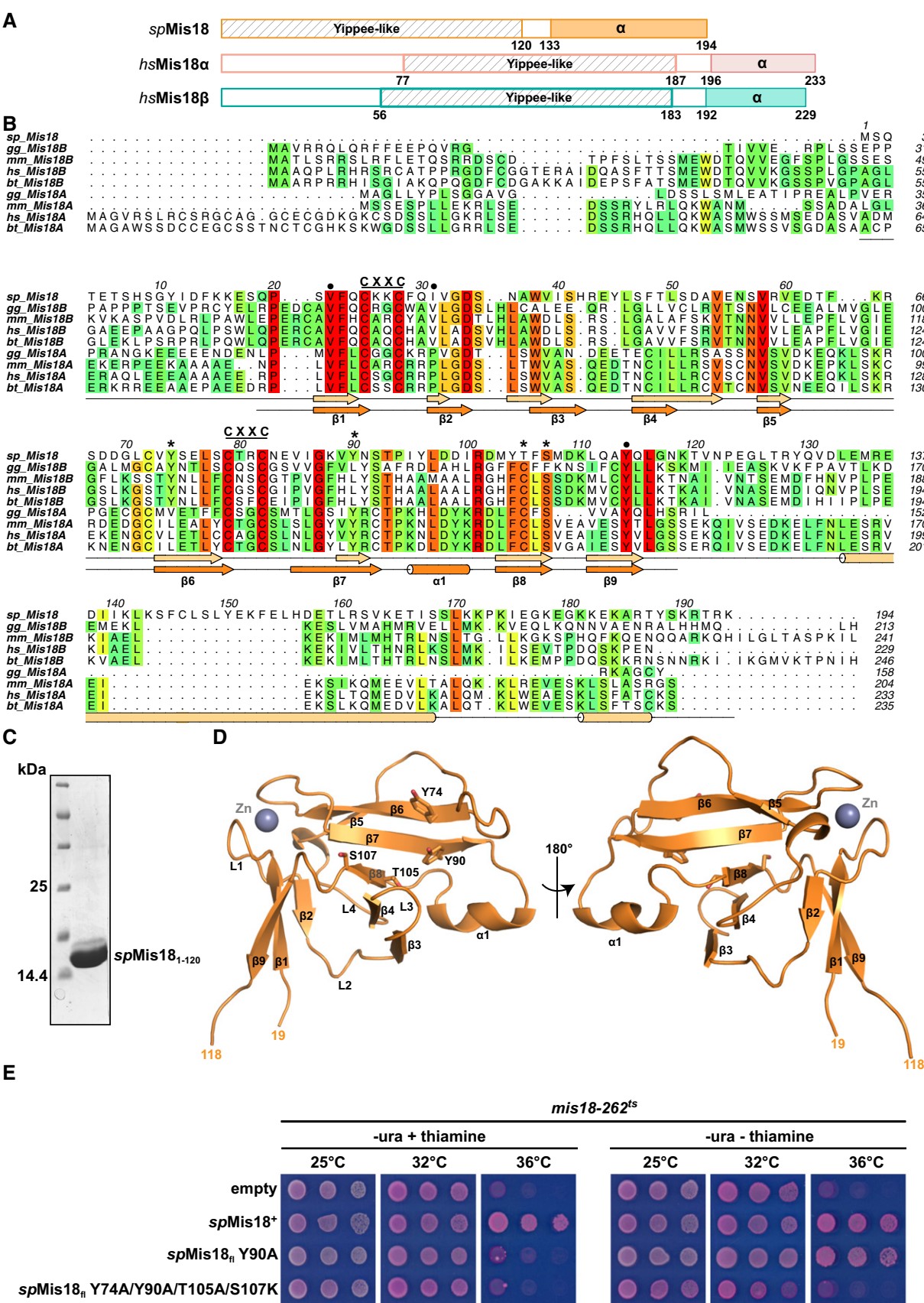

**Figure 1.**

**Table 1. Data collection, phasing, and refinement statistics.**

|  | Dataset I | Dataset II |
|---|---|---|
| Data collection |  |  |
| Space group | P3₁12 | P3₁12 |
| Cell dimensions |  |  |
| *a, b, c* (Å) | 122.18, 122.18, 73.48 | 121.07, 121.07, 73.21 |
| α, β, γ (°) | 90, 90, 120 | 90, 90, 120 |
|  | *Peak* |  |
| Wavelength | 1.2825 | 0.97625 |
| Resolution (Å) | 47.0–3.9 (4.0–3.9) | 104.9–2.6 (2.8–2.6) |
| $R_{merge}$ | 9.5 (98.3) | 6.5 (57.3) |
| I/σI | 15.3 (3.7) | 9.7 (1.8) |
| Completeness (%) | 97.2 (93.9) | 99.9 (99.9) |
| Redundancy | 9.9 (10.3) | 4.1 (4.1) |
| Refinement |  |  |
| Resolution (Å) |  | 39.6–2.6 |
| No. reflections |  | 17,128 |
| $R_{work}/R_{free}$ |  | 22.0/26.0 |
| No. atoms |  |  |
| Protein |  | 2,374 |
| *B*-factors |  |  |
| Protein |  | 98.0 |
| R.m.s deviations |  |  |
| Bond lengths (Å) |  | 0.01 |
| Bond angles (°) |  | 1.75 |
| Ramachandran values |  |  |
| Favored (%) |  | 93 |
| Disallowed (%) |  | 1 |

Values in parentheses are for highest-resolution shell.

Figs 1D and EV1D) failed to complement the loss of *sp*Mis18 function, demonstrating the requirement of this pocket for Mis18 function (Fig 1E).

## *sp*Mis18₁₋₁₂₀ forms a homodimer

The asymmetric unit of the *sp*Mis18₁₋₁₂₀ crystals contained three copies of *sp*Mis18₁₋₁₂₀ assembled in a linear arrangement via two different interfaces (interface I and interface II), resulting in two potential dimeric structures (dimer I and dimer II) (Fig 2A). While the interface stabilizing dimer I is formed by the stacking of loops L5 and L7 of one monomer over their dimeric counterpart, the dimer II interface is formed by the stacking of β-sheet I (Fig 2A). The oligomeric assembly observed in the crystals prompted us to characterize the oligomeric structure of *sp*Mis18₁₋₁₂₀ in solution. The molecular weight of *sp*Mis18₁₋₁₂₀, as measured using SEC-MALS (size exclusion combined with multi-angle light scattering), was 33,338 Da. Given that the calculated molecular weight of an *sp*Mis18₁₋₁₂₀ monomer is 16,160 Da whereas a dimer would be 32,320 Da, this analysis independently confirms that *sp*Mis18₁₋₁₂₀ forms a dimer in solution (Figs 2B and EV2A).

To identify the physiologically relevant dimer, we compared the extent of amino acid conservation of the interface residues and the buried surface area within the dimer I and dimer II interfaces. Dimer I is mainly stabilized by poorly conserved amino acid residues (F64, R66, L71, V73, S92, and I95 with the exception of R101) and involves 1,077 Å² buried surface area. In contrast, dimer II is stabilized by highly conserved residues (V22, I31, Y114, and L116) and has an extensive binding interface (as compared to dimer I) with a 1,431 Å² buried surface area. Based on this observation, we reasoned that dimer II is likely the physiologically relevant dimer. To test the basis of dimer formation, we generated several single point mutations at the dimer II interface (I31A, Y114A, Y114E, V22E) and determined whether they perturb the ability of *sp*Mis18₁₋₁₂₀ to dimerize. In size-exclusion chromatography (Fig 2B), recombinant *sp*Mis18₁₋₁₂₀I31A mutant protein eluted later (Ve = 12.4 ml) than wild-type *sp*Mis18₁₋₁₂₀ (Ve = 11.8 ml), suggesting that this mutant protein forms a smaller entity than wild-type. Further analysis of *sp*Mis18₁₋₁₂₀I31A using SEC-MALS confirmed that it is monomeric with a measured molecular weight of 15,955 Da (calculated molecular weight of a monomer is 16,160 Da) (Figs 2B and EV2A). Consistent with this, the other dimer II interface mutants Y114A, Y114E, and V22E also behaved as smaller entities compared to the wild-type protein when analyzed by native PAGE (Fig EV2B), thus demonstrating that the dimerization of *sp*Mis18₁₋₁₂₀ is mediated via the dimer II interface (Fig 2A).

## Yippee-like globular domains of Mis18 proteins possess an intrinsic ability to form dimers

To test whether Yippee-like globular domains of other Mis18 orthologs also display an intrinsic ability to form dimers, we expressed and purified recombinant fragments of human Mis18α (residues 77–187; *hs*Mis18α₇₇₋₁₈₇) and Mis18β (56–183; *hs*Mis18β₅₆₋₁₈₃) containing their Yippee-like globular domains. SEC and SEC-MALS analyses were conducted to assess their ability to form oligomers. While *hs*Mis18α₇₇₋₁₈₇ eluted as a dimer at 15.5 ml in SEC with a measured molecular weight of 31,254 Da in SEC-MALS (calculated MW of a dimer is 29,324 Da), the corresponding values for *hs*Mis18β₅₆₋₁₈₃ were 17.3 ml and 14,993 Da (calculated MW of a monomer is 13,591 Da), respectively, indicating that *hs*Mis18β₅₆₋₁₈₃ exists as a monomer in solution (Fig 2C). We next tested whether *hs*Mis18α₇₇₋₁₈₇ and *hs*Mis18β₅₆₋₁₈₃ could interact to form a heterodimer, by mixing equimolar quantities of recombinant *hs*Mis18α₇₇₋₁₈₇ and *hs*Mis18β₅₆₋₁₈₃. In SEC, *hs*Mis18β₅₆₋₁₈₃ co-elutes with *hs*Mis18α₇₇₋₁₈₇ at 15.4 ml suggesting that they form a heterodimer (Fig 2C). The molecular weight of this entity as measured by SEC-MALS was 27,476 Da, confirming that *hs*Mis18α and *hs*Mis18β can heterodimerize through their respective Yippee-like domains (Figs 2C and EV2C).

To test whether the mode of dimerization (dimer II mediated) is conserved from fission yeast to humans, we generated a homology model of the human *hs*Mis18α₇₇₋₁₈₇–*hs*Mis18β₅₆₋₁₈₃ heterodimer using the *sp*Mis18₁₋₁₂₀ crystal structure described above as a template (Phyre2 server: http://www.sbg.bio.ic.ac.uk/phyre2/html/page.cgi?id=index) (Fig 2D). Our analysis of the modeled dimer interface (containing Mis18α residues Val 82, Arg 89, Pro 91,

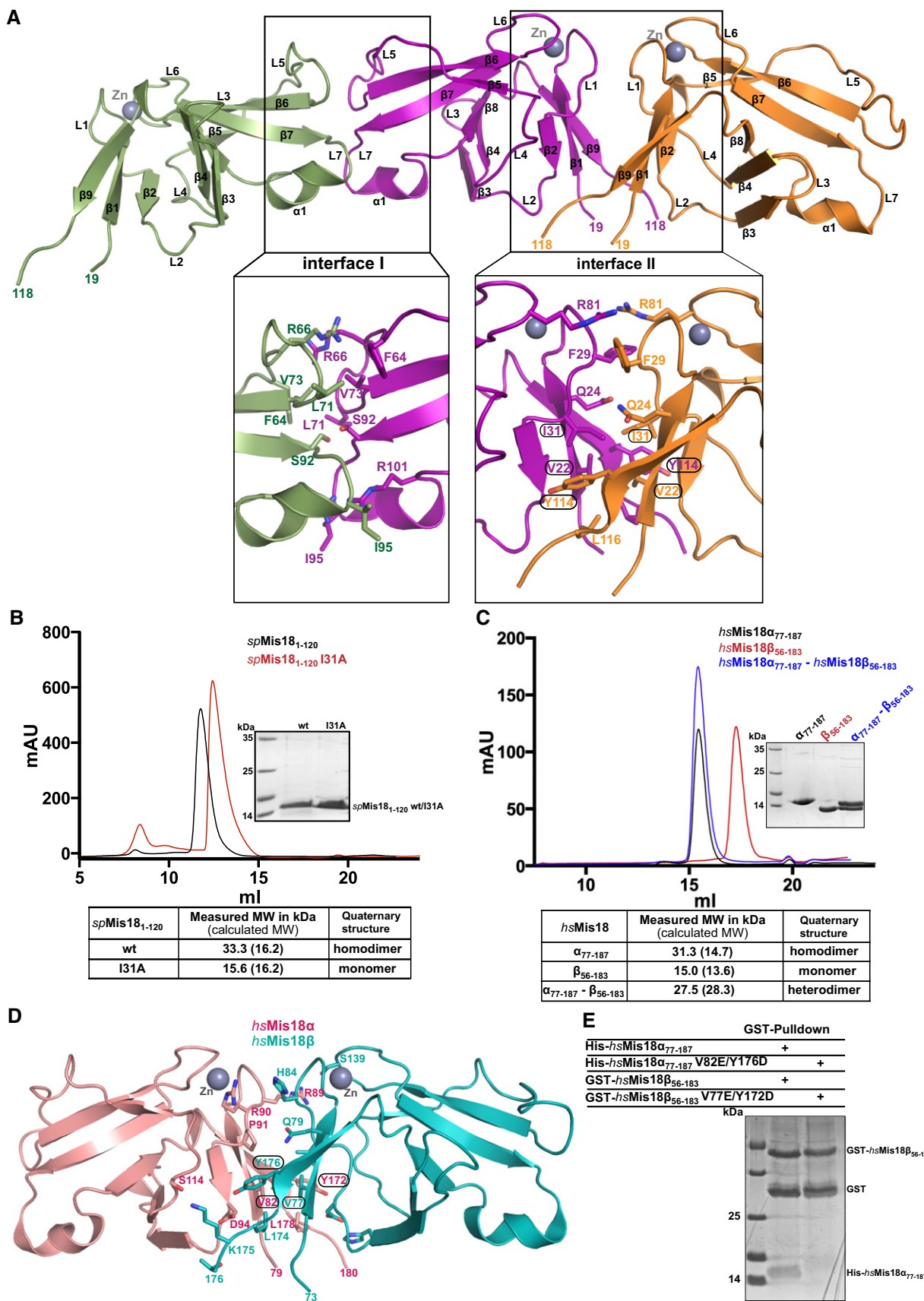

Figure 2.

◄

**Figure 2.  Yippee-like globular domains of Mis18 proteins possess an intrinsic preference to form homo/heterodimers.**

A   Cartoon representation of the assembly of *sp*Mis18 Yippee-like domains (*sp*Mis18$_{1-120}$) as observed in the crystal asymmetric unit. Close-up views highlight the amino acid composition and interactions stabilizing the dimeric arrangements. Interface I is composed of conserved residues and an extensive binding surface as compared to interface I (see text for details), suggesting dimer II to be a physiologically relevant dimer. Residues mutated in this study are highlighted with circles.

B   Size-exclusion chromatography (SEC) and SEC-MALS (SEC combined with multi-angle light scattering) analyses of the recombinant wild-type (wt) and dimer-disrupting mutant (I31A) of *sp*Mis18$_{1-120}$. Measured molecular weights from SEC-MALS confirm that *sp*Mis18$_{1-120}$ wt is a dimer in solution and *sp*Mis18$_{1-120}$I31A is a monomer.

C   Evaluation of the ability of Yippee-like domains of human Mis18 proteins (*hs*Mis18α$_{77-187}$ and *hs*Mis18β$_{56-183}$) to form a heterodimer. SEC and SEC-MALS analyses demonstrate that while *hs*Mis18α$_{77-187}$ elutes at 15.5 ml as a dimer (31.3 kDa), *hs*Mis18β$_{56-183}$ elutes at 17.3 ml as a monomer (15.0 kDa). Purified *hs*Mis18α$_{77-187}$ and *hs*Mis18β$_{56-183}$ when mixed together elute at 15.4 ml as a heterodimer (27.5 kDa).

D   Cartoon representation of the structure of the homology-modeled human Mis18α$_{77-187}$–Mis18β$_{56-183}$ heterodimer using the crystal structure of *sp*Mis18$_{1-120}$ reported here as a template. Residues mutated in this study are highlighted with circles. Modeling was carried out using Phyre2 web server (www.sbg.bio.ic.ac.uk/phyre2/).

E   SDS–PAGE analysis of the GST pull-down assay where wt and dimer-disrupting mutants of *hs*Mis18α$_{77-187}$ and *hs*Mis18β$_{56-183}$ were co-expressed as His- and GST-tagged proteins, respectively, in *E. coli*. While wt GST-*hs*Mis18β$_{56-183}$ showed interaction with wt His-*hs*Mis18α$_{77-187}$, *hs*Mis18 proteins harboring dimer-disrupting mutations, GST-*hs*Mis18β$_{56-183}$V77E/Y72D, and His-*hs*Mis18α$_{77-187}$V82E/Y176D did not show noticeable interaction. The corresponding Ni-NTA pull-downs showing the expression of His-tagged proteins are shown in Fig EV2D.

Asp 94, Tyr 176, and Leu 178, and Mis18β residues Val 77, Val 86, His 92, Tyr 172, Leu 174, Lys 175, and Thr 176) did not show any steric clashes and involved 1,387 Å$^2$ buried surface area, similar to the *sp*Mis18$_{1-120}$ dimer II interface. Moreover, mutations at the dimer interface (Mis18α V82E/Y176D and Mis18β V77E/Y172D) were sufficient to perturb the ability of *hs*Mis18α$_{77-187}$ and *hs*Mis18β$_{56-183}$ to form a heterodimer (Figs 2E and EV2D). This confirms that the Yippee-like domains within Mis18 proteins employ a conserved mode of dimerization.

Structural and biochemical analyses of Yippee-like domains from other proteins (Cereblon, RIG-I and MsrB) have so far yielded no direct evidence that they possess an innate tendency to oligomerize *in vitro*. We therefore refer to the Yippee-like domain of Mis18 orthologs as <u>M</u>is <u>e</u>ighteen <u>D</u>imerization <u>i</u>n <u>Y</u>ippee (MeDiY) domain hereafter.

### The C-terminal α-helical domain induces tetramerization of *S. pombe* Mis18

In addition to the N-terminal MeDiY domain, Mis18 orthologs possess a C-terminal α-helical domain *sp*Mis18$_{C-term-α}$ (aa residues 121–end). To test whether *sp*Mis18$_{C-term-α}$ can influence the overall oligomeric state of the protein, we purified recombinant full-length *sp*Mis18 (*sp*Mis18$_{fl}$). Obtaining intact samples of *sp*Mis18$_{fl}$ proved difficult either with or without His/His-GFP tag, as it was sensitive to degradation from the C-terminus (Fig EV3A). The stable partially degraded His-GFP-*sp*Mis18$_{fl}$ when analyzed by SEC-MALS appeared to form a tetramer (Fig 3A). Mass spectrometric peptide sequence coverage analysis of this sample revealed the loss of approximately 20 amino acids at the C-terminus. Close examination of the amino acid sequence revealed the presence of a low-complexity region at the extreme C-terminus (amino acid residues 171–end, 15 out of 20 amino acid residues being Lys/Arg) that is unique to *sp*Mis18 (Fig EV3B). We therefore expressed and purified recombinant *sp*Mis18 with a C-terminal truncation (*sp*Mis18ΔC; amino acids 1–168; Fig EV3C). SEC-MALS analysis of *sp*Mis18ΔC revealed that it predominantly formed a tetramer (Fig 3B) and demonstrated a role for the α-helical region downstream of the MeDiY domain (residues 121–168; Fig 1B) in *sp*Mis18 tetramerization. Since this α-helical region is structurally conserved (based on secondary structure prediction; Fig 1B) between Mis18 proteins from evolutionarily distant eukaryotes, we propose that Mis18 oligomerization (either homo or hetero) mediated through the C-terminus is also likely to be highly conserved.

### MeDiY and the C-terminal α-helical domain are independent structural modules

To obtain further insights into the overall architecture of the *sp*Mis18 oligomer, we tested whether *sp*Mis18$_{MeDiY}$ and *sp*Mis18$_{C-term-α}$ could interact with each other, or whether they existed as structurally independent modules. His-*sp*Mis18$_{MeDiY}$ and His-GFP-*sp*Mis18$_{C-term-α}$ were purified individually and analyzed for complex formation using size-exclusion chromatography. His-GFP-*sp*Mis18$_{C-term-α}$ and His-*sp*Mis18$_{MeDiY}$ eluted separately at distinct elution volumes (12.3 and 15.3 ml, respectively). This demonstrates that the *sp*Mis18$_{MeDiY}$ domain alone is unable to associate with *sp*Mis18$_{C-term-α}$ (Figs 3C and EV3D).

We next tested whether *sp*Mis18$_{C-term-α}$ can self-associate to form oligomers. SEC-MALS analysis of His-GFP-*sp*Mis18$_{C-term-α}$ (untagged *sp*Mis18$_{C-term-α}$ (8,775 Da) is too small for accurate mass detection) revealed that it is a homotrimer with a measured molecular weight of 119,886 Da (theoretical molecular weight of a monomer is 38,337 Da while that of a homotrimer would be 115,011 Da) (Fig 3D). We therefore conclude that *sp*Mis18 has two structurally independent domains, and they each possess the ability to homo-oligomerize.

### Oligomerization of *sp*Mis18 via the MeDiY domain is required for its function

We next tested whether *sp*Mis18 dimerization is functionally important *in vivo*. Co-immunoprecipitation assays were performed on *S. pombe* cells expressing endogenous *sp*Mis18$_{fl}$ with a C-terminal 8xMyc tag, and *sp*Mis18$_{fl}$ or *sp*Mis18$_{fl}$I31A expressed ectopically as a C-terminally GFP-tagged fusion protein. Endogenous wild-type *sp*Mis18$_{fl}$-Myc associated efficiently with ectopic wild-type *sp*Mis18$_{fl}$-GFP, confirming the interaction of *sp*Mis18$_{fl}$ with itself *in vivo* (Fig 4A). Such self-association of wild-type *sp*Mis18 has also been demonstrated previously through yeast two-hybrid assays [13,15]. In contrast, the *sp*Mis18$_{fl}$I31A dimerization mutant showed significantly reduced association with endogenous *sp*Mis18$_{fl}$-Myc. Additionally, *sp*Mis18$_{MeDiY}$ alone was sufficient to associate with *sp*Mis18$_{fl}$-Myc (Fig 4A). This highlights the role of *sp*Mis18$_{MeDiY}$ in the overall oligomeric state of *sp*Mis18 *in vivo*.

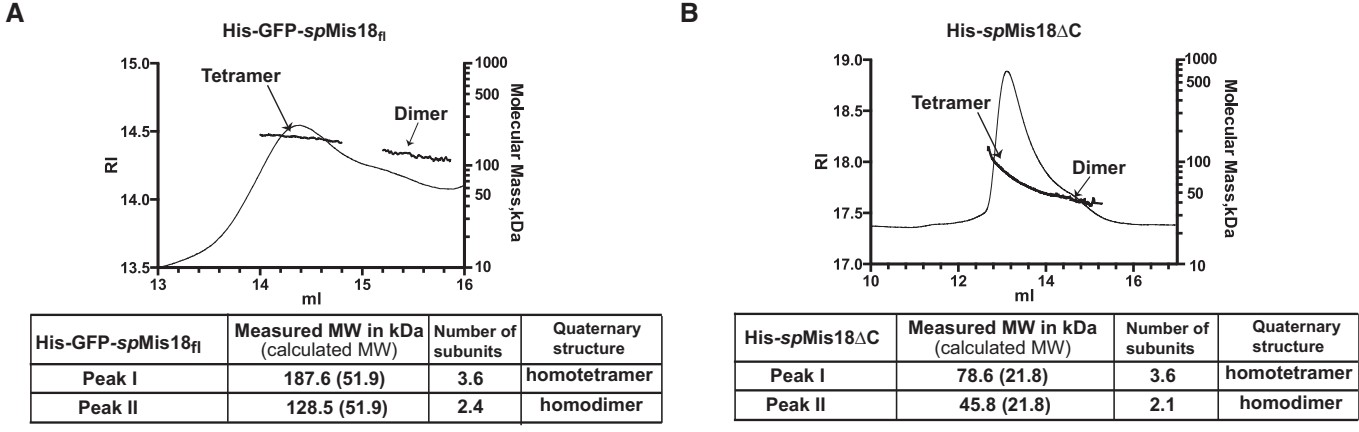

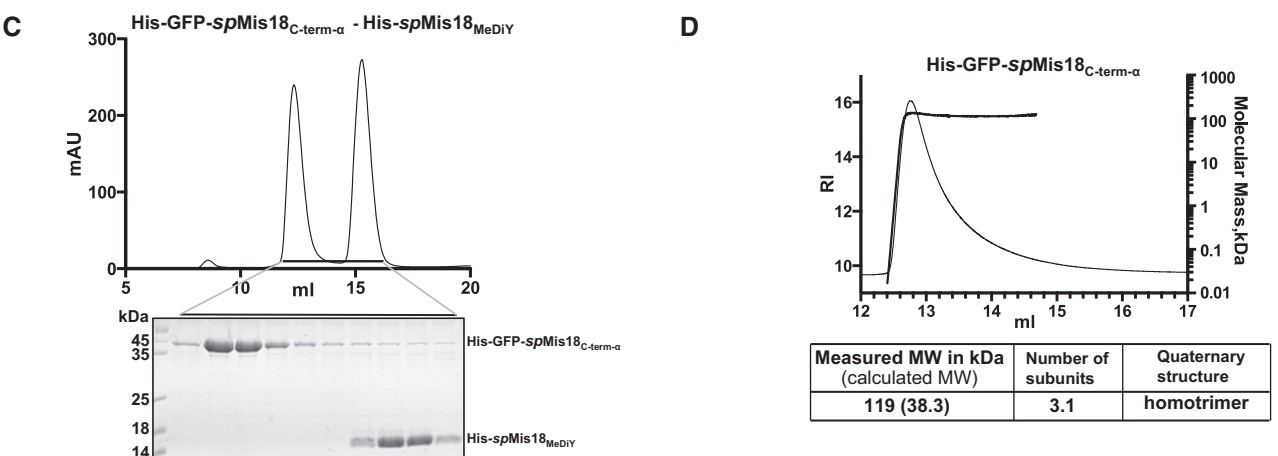

**Figure 3.** *sp*Mis18$_{fl}$ forms a tetramer *in vitro*.

A, B SEC-MALS analysis of His-GFP-*sp*Mis18$_{fl}$ and His-*sp*Mis18ΔC, respectively. Refractive index (RI, left *y*-axis) and molecular mass (kDa, right *y*-axis) profiles show that both His-GFP-*sp*Mis18$_{fl}$ (predicted MW of a monomer is 51.9 kDa) and His-*sp*Mis18ΔC (predicted MW of a monomer 21.8 kDa) predominantly exist as tetramers (187.6 and 78.6 kDa, respectively) with a small population of dimer (128.5 and 45.8 kDa, respectively).

C SEC profile of a sample containing an equimolar mixture of purified His-GFP-*sp*Mis18$_{C-term-α}$ and His-*sp*Mis18$_{MeDiY}$ (upper panel) and SDS–PAGE analysis of SEC fractions (bottom panel). His-GFP-*sp*Mis18$_{C-term-α}$ and His-*sp*Mis18$_{MeDiY}$ eluted separately at 12.3 and 15.3 ml, respectively, demonstrating the inability of these domains to interact with each other.

D SEC-MALS analysis of His-GFP-*sp*Mis18$_{C-term-α}$ shows that it exists as a homotrimer with a measured molecular weight of 119.8 kDa (theoretically calculated MW of a monomer is 38.3 kDa).

To test whether *sp*Mis18$_{MeDiY}$-mediated oligomerization is essential for *sp*Mis18 function, we performed genetic complementation assays in which we evaluated the ability of dimerization mutants I31A and Y114A (in the context of both *sp*Mis18$_{fl}$ and *sp*Mis18$_{MeDiY}$), to rescue the temperature-sensitive growth phenotype of *mis18-818* (T49A) and *mis18-262* (G117D) cells *in vivo* (Fig 4B and C) [10]. Dimer-disrupting mutants *sp*Mis18$_{fl}$I31A and *sp*Mis18$_{fl}$Y114A failed to rescue the viability defect at restrictive temperature in both *mis18-262* and *mis18-818* cells. Expression of *sp*Mis18$_{MeDiY}$ alone, while failing to complement the loss of *sp*Mis18 function at 36°C, reproducibly conferred a dominant-negative effect on growth in both *mis18-262* and *mis18-818* cells at 32°C. This inhibitory effect on growth of *mis18-262* and *mis18-818* cells depended on the ability of *sp*Mis18$_{MeDiY}$ to dimerize, as the *sp*Mis18$_{MeDiY}$I31A and Y114A mutations caused no such negative influence on growth (Fig 4B and C). We conclude that dimerization of *sp*Mis18 via the MeDiY domain is crucial for *sp*Mis18 function *in vivo*.

## Centromeric localization of *sp*Mis18 depends on MeDiY-mediated dimerization

*sp*Mis18 is known to associate with centromeres, where it is required for the incorporation of CENP-A$^{Cnp1}$ [10]. Chromatin immunoprecipitation analyses were performed to determine whether dimerization mediated by the MeDiY domain is required for the association of *sp*Mis18$_{fl}$ with centromeres. Disruption of *sp*Mis18 MeDiY-mediated dimerization through *sp*Mis18$_{fl}$I31A or *sp*Mis18$_{fl}$Y114A mutations resulted in reduced association of ectopically expressed *sp*Mis18$_{fl}$-GFP with centromeres in wild-type cells (Fig 4D), although no significant defects in subcellular localization of *sp*Mis18$_{fl}$-GFP were observed (Fig EV4A). Additionally, cell growth and CENP-A$^{Cnp1}$ association with centromeres were essentially unaffected in these cells (Fig EV4B and C). Thus, MeDiY domain-mediated dimerization ensures optimal *sp*Mis18 association with centromeres.

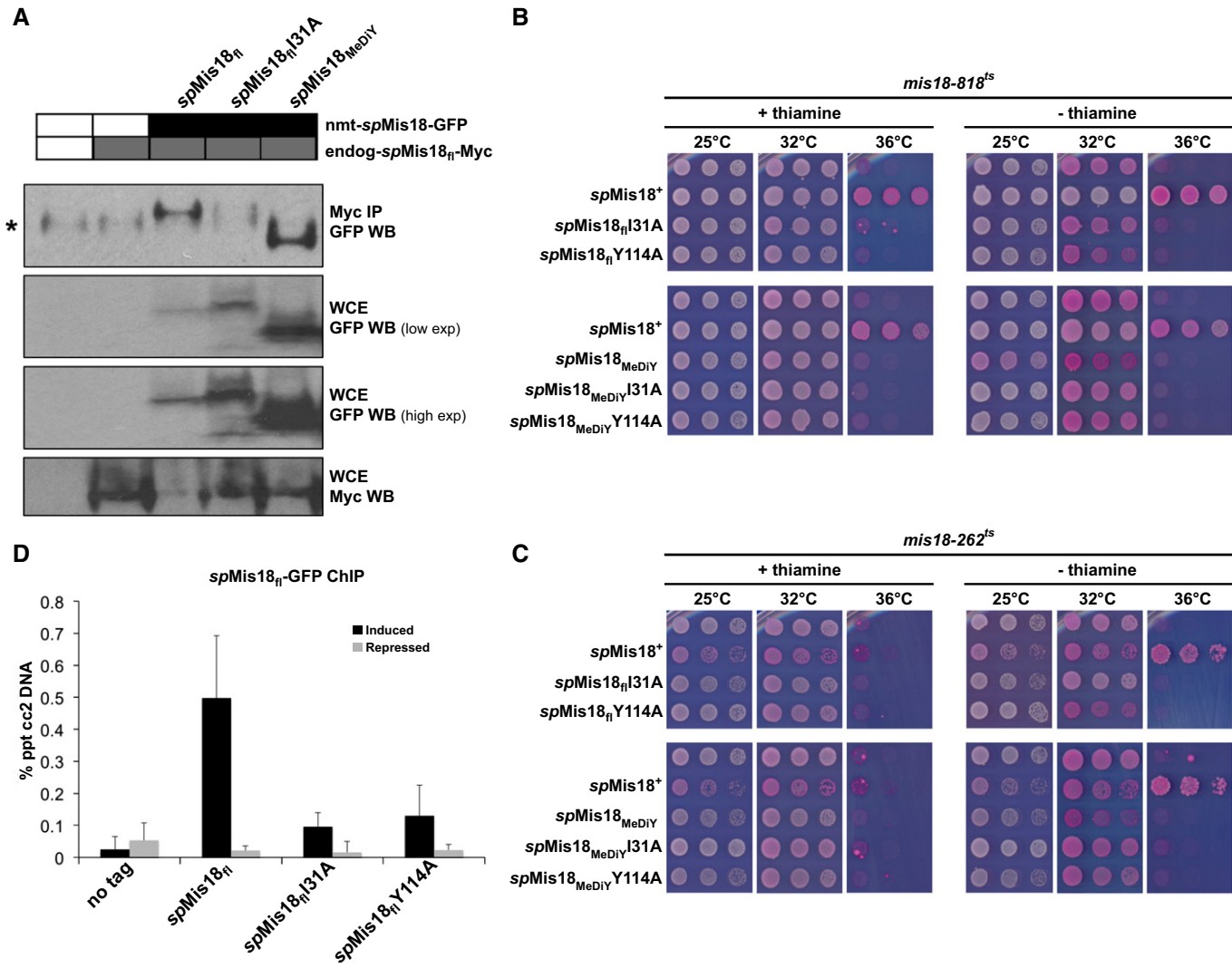

**Figure 4.  Dimerization mediated by the MeDiY domain promotes *sp*Mis18 function and centromere association.**

A     *sp*Mis18<sub>fl</sub>-Myc co-immunoprecipitates with both GFP-tagged *sp*Mis18<sub>fl</sub> and *sp*Mis18<sub>MeDiY</sub>, but shows reduced association when MeDiY-mediated dimerization is disrupted in *sp*Mis18<sub>fl</sub>I31A. The asterisk (*) in the top panel denotes the IgG heavy chain.

B, C   Dimer II interface mutations I31A and Y114A affect the ability of ectopically expressed *sp*Mis18<sub>fl</sub> to rescue the temperature sensitivity of *mis18-818* and *mis18-262* cells, while expression of *sp*Mis18<sub>MeDiY</sub> alone confers a dominant-negative effect on growth in a MeDiY dimerization-dependent manner. Fivefold serial dilutions of cells expressing the indicated *sp*Mis18 constructs integrated at the *leu1* locus in the genome, spotted on complete PMG + phloxine B media supplemented with (repressed) or without (expressed) thiamine, and incubated at the indicated temperatures; dead cells stain dark pink.

D     Mutations that disrupt MeDiY dimerization lead to reduced levels of *sp*Mis18<sub>fl</sub> association with centromeres. qChIP analyses of *sp*Mis18<sub>fl</sub>-GFP association with centromere 2 (cc2) in the indicated strains when grown in complete PMG media supplemented with (repressed) or without (expressed) thiamine. Error bars represent standard deviation between at least three biological replicates.

Centromeric association of the Mis18 complex is one of the early key steps involved in the establishment of CENP-A nucleosomes at centromeres [17]. Although some of the components of the Mis18 complex show variations among different species (particularly the absence of Mis18BP1/KNL2 in fission yeast), the function of the Mis18 complex in targeting HJURP to centromeres is well conserved [14,15,26–28]. The molecular mechanism by which the Mis18 complex is targeted to centromeres, however, remains to be determined, and any structural insights into the molecular architecture of the Mis18 complex and/or its components have the potential to advance our understanding of centromere establishment.

The Mis18 complex is critical for CENP-A deposition at centromeres; however, Mis18 is poorly characterized in terms of its structure and function. Here, we show that fission yeast Mis18 and its human isoforms, Mis18α and Mis18β, possess two structurally distinct domains: an N-terminal "Yippee-like" globular domain and a C-terminal α-helical domain. Structural alignments revealed unambiguous structural homology between the N-terminal "Yippee-like" globular domain and proteins of diverse function such as Cereblon [21], RIG-I [22], MSS4 [23], and MsrB [24], as well as identified a conserved substrate-binding pocket in Mis18 required for its function. However, predicting the

identity and nature of Mis18 binding partner(s) seems non-trivial, as the amino acids that make up the binding pocket and their spatial orientation are markedly different among the various homologous proteins.

A key finding of our analyses is the ability of the "Yippee-like" globular domain (MeDiY) of *S. pombe* and human Mis18 proteins to dimerize. To date, none of the other related domains are known to form dimers. Interestingly, *S. pombe* Mis18 and also human Mis18 [29], in the presence of Mis18$_{C\text{-term-}\alpha}$, forms a tetramer. Notably, ectopic expression of the MeDiY domain alone, while failing to complement the loss of *sp*Mis18 function conferred a dominant-negative effect on the growth of *mis18* mutants, implicating both MeDiY and C-terminal α-helical domains in ensuring the required oligomeric structure for function. Specifically, perturbing the dimeric interface of the MeDiY domain affects the centromere localization of *sp*Mis18. Identifying the binding partners of MeDiY and determining how (and if) Mis18 oligomerization can influence substrate recognition are important questions to be addressed in the future to unravel the molecular mechanisms that mediate centromere localization and function of the Mis18 complex. A previous study has shown that HJURP dimerization is required for stable deposition of CENP-A nucleosomes at centromeres and implicated a direct role for HJURP dimerization in forming octameric CENP-A nucleosomes [30]. These observations highlight protein oligomerization-mediated regulation as an emerging regulatory theme for the inherently complex process of centromere establishment and maintenance.

# Materials and Methods

### Expression and purification of recombinant *S. pombe* proteins

*Schizosaccharomyces pombe* Mis18$_{fl}$, Mis18$_{MeDiY}$, Mis18ΔC, and Mis18$_{C\text{-term-}\alpha}$ codon-optimized sequences (GeneArt) were cloned into pEC-K-3C-His or 9GFP (Addgene) LIC vectors with N-terminal His or His-GFP tags cleavable with either a 3C or TEV site, respectively. Mutations were introduced using QuikChange site-directed mutagenesis protocol (Stratagene). All proteins were expressed using *E. coli* BL21 Gold; His-*sp*Mis18$_{MeDiY}$ was grown in Super Broth and induced for 6 h at 25°C using 0.3 mM IPTG. His-*sp*Mis18ΔC was grown in Super Broth, while His-GFP *sp*Mis18$_{C\text{-term-}\alpha}$ and His-GFP-*sp*Mis18$_{fl}$ were grown in 2× TY before inducing for 16 h at 18°C by adding 0.3 mM IPTG.

All proteins were lysed via sonication in a lysis buffer containing 20 mM Tris pH 8 (or 8.5 for His-*sp*Mis18$_{MeDiY}$), 35 mM imidazole and 2 mM BME supplemented with 10 μg/ml DNase, 1 mM PMSF, and complete EDTA-free (Roche). The following NaCl concentrations were used in the lysis buffer: 500 mM for His-*sp*Mis18$_{MeDiY}$, 50 mM for His-*sp*Mis18ΔC, and 100 mM for His-GFP-*sp*Mis18$_{C\text{-term-}\alpha}$ or His-GFP-*sp*Mis18$_{fl}$. After clarification, His-GFP-*sp*Mis18$_{C\text{-term-}\alpha}$, His-GFP-*sp*Mis18$_{fl}$, and His-*sp*Mis18$_{MeDiY}$ proteins were purified using a 5-ml HisTrap HP column (GE Healthcare), while His-*sp*Mis18ΔC was purified by batch mode using HisPur Ni-NTA resin (Thermo Scientific). Resin was washed with lysis buffer, and then, His-*sp*Mis18$_{1–120,}$ His-GFP-*sp*Mis18$_{C\text{-term-}\alpha}$ or His-GFP-*sp*Mis18$_{fl}$ received additional washes with 20 mM Tris pH 8/8.5, 35 mM

imidazole, 50 mM KCl, 10 mM MgCl$_2$, 2 mM ATP, and 2 mM BME. The following NaCl concentrations were used: 500 mM for His-*sp*Mis18$_{MeDiY}$ or 1 M for His-GFP-*sp*Mis18$_{C\text{-term-}\alpha}$ and His-GFP-*sp*Mis18$_{fl}$. Proteins were finally washed in lysis buffer and then eluted with 20 mM Tris pH 8/8.5, 500 mM imidazole, and 2 mM BME. The following NaCl concentrations were used in the elution buffer: 500 mM for His-*sp*Mis18$_{MeDiY}$, His-*sp*Mis18ΔC, and His-GFP-*sp*Mis18$_{C\text{-term-}\alpha}$, and 100 mM NaCl for His-GFP-*sp*Mis18$_{fl}$. Eluted His-*sp*Mis18$_{1–120}$ was used directly for crystallization trial, while all other proteins were subjected to size-exclusion chromatography. His-*sp*Mis18ΔC was loaded onto Superdex 200 Hi-load 16/600 (GE Healthcare) equilibrated with 20 mM Tris pH 8, 400 mM NaCl, and 1 mM TCEP. Appropriate fractions were pooled and injected onto a Superdex 200 increase 10/300 column (GE Healthcare) equilibrated with 20 mM Tris pH 8, 50 mM NaCl, and 4 mM DTT. His-GFP-*sp*Mis18$_{C\text{-term-}\alpha}$ was applied to a Superdex 200 increase 10/300 column equilibrated with 20 mM Tris pH 8, 150 mM NaCl, and 2 mM DTT. His-GFP-*sp*Mis18$_{fl}$ was applied to a Superose 6 10/300 column (GE Healthcare) equilibrated with 20 mM Tris pH 8, 300 mM NaCl, and 2 mM DTT. Fractions were analyzed on SDS–PAGE and stained with Coomassie blue.

### Expression and purification of recombinant human proteins

Human Mis18α$_{fl}$, Mis18α$_{77–187}$, Mis18β$_{fl}$, and Mis18β$_{56–183}$ codon-optimized sequences (GeneArt) were cloned into pEC-K-3C-His LIC vector or pGEX-6P-1 (GE Healthcare). Human proteins were expressed separately or co-expressed using pGEX-6P-1 and pEC-K-3C-His in BL21 Gold in a similar manner to *sp*Mis18ΔC. His-*hs*Mis18α$_{77–187}$ was lysed via sonication in 20 mM Tris pH 8, 500 mM NaCl, 35 mM imidazole, and 4 mM BME with 10 μg/ml DNase, 1 mM PMSF, and complete EDTA-free. Clarified lysate was loaded onto a 5-ml HisTrap HP column and washed with lysis buffer and then with 20 mM Tris pH 8, 1 M NaCl, 35 mM imidazole, 50 mM KCl, 10 mM MgCl$_2$, 2 mM ATP, and 4 mM BME and re-equilibrated in lysis buffer before eluting with 20 mM Tris pH 8, 500 mM NaCl, 500 mM imidazole, and 4 mM BME. GST-*hs*Mis18β$_{56–183}$ was sonicated in 20 mM Tris pH 8, 500 mM NaCl, and 4 mM DTT with 10 μg/ml DNase, 1 mM PMSF, and complete EDTA-free. Cleared lysate was loaded on 12 ml glutathione–Sepharose (GE Healthcare) in batch mode and washed in lysis buffer and then with 20 mM Tris pH 8, 1 M NaCl, 50 mM KCl, 10 mM MgCl$_2$, 2 mM ATP, and 4 mM DTT followed by lysis buffer. 3C protease was added to cleave the tag on the beads overnight and untagged protein collected. To obtain the *hs*Mis18α$_{77–187}$/Mis18β$_{56–183}$ complex, proteins were co-expressed and purified as described for GST-*hs*Mis18β$_{56–183}$. All proteins were separated on Superose 6 10/300 pre-equilibrated with 20 mM Tris pH 8, 200 mM NaCl, and 5 mM DTT.

### His/GST pull-down assays

To test for protein–protein interactions, purified His-tagged *sp*Mis18$_{MeDiY}$ and His-GFP-tagged *sp*Mis18$_{C\text{-term-}\alpha}$ proteins were mixed and applied to a Superdex 200 increase 10/300 column equilibrated with 20 mM Tris pH 8, 150 mM NaCl, and 2 mM DTT and fractions analyzed via SDS–PAGE. To test for interaction of human *hs*Mis18α and β, 10 ml of culture of co-expressed proteins was

           © 2016 The Authors

grown and sonicated in 20 mM Tris pH 8, 100 mM NaCl, and 2 mM BME. After clarification, supernatants were split and 35 mM imidazole added to half before incubation for 1 h with either glutathione–Sepharose or HisPur Ni-NTA resin and then washed with buffer. Purified proteins were run on SDS–PAGE.

### Native PAGE

12 % native PAGE gels (22 mM Tris pH 8.8, 12 % acrylamide with stacking 320 mM Tris pH 8.8, 4 % acrylamide) were run in 25 mM Tris and 192 mM glycine at 150 V for 30 mins. Proteins in 1× sample buffer (31.25 mM Tris pH 6.8, 12.5 % glycerol, 0.5 % bromophenol blue) and Native Marker (Life Technologies) were loaded and then run at 150 V for 3 h.

### Crystallization and data collection

Crystallization trials were performed using a nanoliter crystallization robot at the Edinburgh crystallization facility. Crystals were grown by vapor diffusion method. Diffraction quality crystals were obtained using well buffer containing 0.2 M ammonium chloride/formate/acetate/phosphate and 20 % PEG 3350 (with the measured pH of the solution in the range of 6.2–8). 15–20 mg/ml protein sample was mixed with the well buffer in a 1:1 ratio. Crystals were briefly transferred to cryoprotectant solution (crystallization solutions were supplemented with glycerol or ethylene glycol to a final concentration 25 %) before flash cooling in liquid nitrogen. The crystals diffracted to 2.6 Å resolution at the MX beamlines of the Diamond Light Source (Table 1).

### Crystal structure solution and refinement

The structure of $sp$Mis18$_{MeDiY}$ was determined by the single anomalous dispersion (SAD) method using the anomalous signal of intrinsically bound $Zn^{2+}$ ion. Data were processed using XIA2 and scaled with SCALA of CCP4 [31]. SAD phasing and the calculation of the initial map were performed using phenix.autosol from the PHENIX suite of programs [32]. The model was built by iterative rounds of manual building with COOT [33], and refinement was done using Refmac5 of CCP4. Data collection, phasing, and refinement statistics are shown in Table 1.

### SEC-MALS

Size-exclusion chromatography coupled to multi-angle light scattering (SEC-MALS) was performed at the Edinburgh Protein Production Facility at room temperature using a ÄKTA FPLC. 100 µl of protein at 1 mg/ml was loaded onto either Superose 6 10/300, Superdex 75 10/300, or Superdex 200 10/300 columns (GE Healthcare) pre-equilibrated with 50 mM HEPES pH 8, 150 mM NaCl, and 5 mM DTT for His-$sp$Mis18$_{MeDiY}$, His-GFP-$sp$Mis18$_{C\text{-term-}\alpha}$, and $hs$Mis18α$_{77–187}$/$hs$Mis18β$_{56–183}$, or 50 mM HEPES pH 8, 50 mM NaCl, and 5 mM DTT for His-$sp$Mis18ΔC or 50 mM HEPES pH 8, 300 mM NaCl, and 1 mM TCEP for His-GFP $sp$Mis18$_{fl}$. A Mini-DAWN in-line detector (Wyatt Technology) was used to measure MALS, while a Viscotek RI Detector (Wyatt Technology) was used to detect refractive index. Data were analyzed using ASTRA™ software (Wyatt Technology).

### Plasmids and *S. pombe* strains

For *in vivo* assays in *S. pombe*, $sp$Mis18 cDNA (wt, I31A, Y114A, Y90A or Y74A Y90A T105A S107K in the context of Mis18$_{fl}$ or Mis18$_{MeDiY}$) was PCR-amplified and cloned into the pDUAL-GFH41 vector, which allows for expression of C-terminally GFP-tagged $sp$Mis18 (wt or mutants) under the control of the medium-strength *nmt41* promoter that is induced in the absence of thiamine in culture media [34]. $sp$Mis18 constructs cloned into the pDUAL-GFH41 vector were then either transformed as such and selected in the absence of uracil in growth media, or integrated into the genome at the *leu1* locus, in wt, *mis18-818,* or *mis18-262* cells. Genotypes of *S. pombe* strains used in this study are listed in Table EV1.

### Genetic complementation assays

Fivefold serial dilutions of *mis18-262, mis18-818,* or wt cells expressing GFP-tagged $sp$Mis18$_{fl}$ or $sp$Mis18$_{MeDiY}$ (wt or mutants) either from an ectopic pDUAL-GFH41 plasmid (selected for in media lacking uracil; Fig 1E) or from the *leu1* locus (integrated in the genome; Figs 4B and C and EV4B), were spotted onto PMG media containing phloxine B supplemented with or without thiamine and incubated at the indicated temperatures for 3–5 days.

### Co-immunoprecipitation and Western analyses

For co-immunoprecipitation experiments, cells expressing endogenous $sp$Mis18$_{fl}$-Myc and ectopic $sp$Mis18-GFP (wt or mutants; from the *leu1* locus) were cultured in complete PMG media lacking thiamine for 21 h and processed as previously described [11]. Immunoprecipitation was performed using rabbit anti-myc antibody A14 (Santa Cruz Biotech) and Western analyses using monoclonal anti-GFP (Roche) or anti-myc 9B11 (Cell Signaling).

### Quantitative chromatin immunoprecipitation

Anti-GFP ChIP, anti-CENP-A$^{Cnp1}$ ChIP, and real-time PCR analyses were performed as previously described [11], on wild-type *S. pombe* cells expressing ectopic GFP-tagged $sp$Mis18$_{fl}$ (wt or mutants; from the *leu1* locus) cultured in complete PMG media lacking thiamine for 21 h.

### Cytology

Immunolocalization and microscopy were performed as previously described [11] on wild-type *S. pombe* cells expressing ectopic GFP-tagged $sp$Mis18$_{fl}$ (wt or mutants; from the *leu1* locus) cultured in complete PMG media lacking thiamine for 21 h.

### Data deposition

Atomic coordinates of the structure and structure factors are deposited in the RCSB protein data bank (www.rcsb.org) with accession code 5HJ0.

**Expanded View** for this article is available online.

## Acknowledgements

We thank D. Foltz for sharing data prior to publication. We thank the staff of Diamond Light Source for assistance during data collection and staff of Edinburgh protein production facility, and particularly M. Wear for his help with SEC-MALS data generation and analysis. We thank C. Ponting and L. Sanchez-Pulido for the initial bioinformatics analysis of the Mis18 Yippee-like domain. We also thank L. Mackay for his help with the MS-based protein sequence coverage analysis of Mis18. We are grateful to M. Yanagida and the National Bioresource Project (YGRC) Japan for *S. pombe* strains, and P. Heun, and members of Jeyaprakash and Allshire laboratory for discussions. We also thank W. Earnshaw, A. Cook, K. Hardwick, J. Welburn, A. Pidoux, and M. A. Abad Fernandez for the critical reading of the manuscript. The Wellcome Trust generously supported this work through a Wellcome Trust Career Development Grant to AAJ (095822), a Principal Research Fellowship to RCA (095021 & 065061), a Wellcome Trust Centre Core Grant (092076), an instrument grant (091020), and the Wellcome Trust-University of Edinburgh Institutional Strategic Support Fund. The European Commission supported this work through a career integration grant to AAJ and through an EpiGeneSys Network of Excellence grant to RCA (HEALTH-F4-2010-257082. LS was supported by an EC FP7 Marie Curie International Incoming Fellowship (PIIF-GA-2010-275280) and an EMBO Long Term Fellowship (ALTF 1491-2010). RB was supported by a Wellcome Trust 4-year PhD program.

## Author contributions

LS, BM-P, RB, FS, RK-S, GR, and AAJ performed experiments and analyzed data. LS, BM-P, RCA, and AAJ designed experiments, analyzed data, and wrote the manuscript.

## Conflict of interest

The authors declare that they have no conflict of interest.

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
