## [Review Process File · EMBO Reports]

Manuscript EMBO-2015-41520

Centromere Localization and Function of Mis18 Requires 'Yippee-like' Domain-Mediated Oligomerization

Lakxmi Subramanian, Bethan Medina-Pritchard, Rachael Barton, Frances Spiller, Raghavendran Kulasegaran-Shylini, Guoda Radaviciute, Robin C: Allshire, A. Arockia Jeyaprakash

Corresponding author: A. Arockia Jeyaprakash, University of Edinburgh

Review timeline:

Submission date:	06 October 2015
Editorial Decision:	09 November 2015
Revision received:	19 January 2016
Accepted:	03 February 2016

Editor: Achim Breiling/Martina Rembold

Transaction Report:

1st Editorial Decision

09 November 2015

Thank you for the submission of your research manuscript to EMBO reports. We have now received the full set of referee reports that is copied below.

As you will see, all three referees acknowledge that the findings are interesting. Referee 1 recommends several changes to the text and figure legends to improve the clarity. Referee 3, among other points, suggests experiments to better demonstrate the mislocalization of the Mis18 dimer interface mutants (point 3) and to analyze the CENP-A deposition in cells expressing the I31A or Y114A mutants (point 4).

Given these constructive comments, we would like to invite you to revise your manuscript with the understanding that the referee concerns must be fully addressed and their suggestions taken on board, with the exception of point 2 of referee 3, which seems beyond the scope of the current study and is not mandatory. Please address all referee concerns in a complete point-by-point response. Acceptance of the manuscript will depend on a positive outcome of a second round of review. It is EMBO reports policy to allow a single round of revision only and acceptance or rejection of the manuscript will therefore depend on the completeness of your responses included in the next, final version of the manuscript.

Revised manuscripts should be submitted within three months of a request for revision; they will otherwise be treated as new submissions. Please contact us if a 3-months time frame is not sufficient for the revisions so that we can discuss the revisions further. For short reports, the revised manuscript should not exceed 35,000 characters (including spaces and references) and 5 main plus 5 expanded view figures. The results and discussion section must further be combined, which will

help to shorten the manuscript text by eliminating some redundancy that is inevitable when discussing the same experiments twice.

Please deposit your structural data in an appropriate database (PDB or NDB) and indicate the deposition and corresponding access number in the manuscript.

We now strongly encourage the publication of original source data with the aim of making primary data more accessible and transparent to the reader. The source data will be published in a separate source data file online along with the accepted manuscript and will be linked to the relevant figure. If you would like to use this opportunity, please submit the source data (for example scans of entire gels or blots, data points of graphs in an excel sheet, additional images, etc.) of your key experiments together with the revised manuscript. Please include size markers for scans of entire gels, label the scans with figure and panel number, and send one PDF file per figure or per figure panel.

As part of the EMBO publication's Transparent Editorial Process, EMBO reports publishes online a Review Process File to accompany accepted manuscripts. This File will be published in conjunction with your paper and will include the referee reports, your point-by-point response and all pertinent correspondence relating to the manuscript.

I look forward to seeing a revised version of your manuscript when it is ready. Please let me know if you have questions or comments regarding the revision.

REFeree REPORTS

Referee #1:

Mis18 and the Mis18 complex are key regulators of centromere specification from yeast to vertebrates. However, little is known about the precise function of this protein family. Through a combination of structural biology and yeast genetics, Subramanian et al. identify a conserved portion of the Mis18 Yippee-like domain that mediates dimerization and oligomerization in vitro and in vivo in *S. pombe*. Additionally, they show an essential role for a 'binding pocket' region present in the Yippee domain. The authors show that human Mis18a, and Mis18b also form hetero-dimers, demonstrating conservation of the role of the Yippee-like domain in mediating dimerization. The specific residues within the Yippee-like domain mediating the dimerization are identified. Mutations in these residues fail to rescue lethality of temperature-sensitive Mis18 mutants. Additionally, dimerization defective Mis18 mutants are unable to bind centromeric chromatin.

The authors did not identify the target of the Mis18 substrate-binding pocket, but this is understandably challenging and beyond the scope of the present work.

The minor comments below focus primarily on improving the clarity of the paper and should be easily addressed during revision.

-The abstract opens with a sentence that is only relevant to humans. I suggest specifying that or rewording to be either more general or specific to *pombe*, which is the main system relevant here.

-Introduction, page 4 near the top: "Exclusion of the Mis18 complex and HJURPScm3 from centromeres during mitosis provides an opportunity for the CENP-ACnp1 loading cycle to reset"; what are the authors trying to say?

-Same page: "As in *S. pombe*, the human Mis18 complex is required for HJURP recruitment to centromeres, where it deposits...", the wording here suggests Mis18 deposits CENP-A rather than HJURP.

-Bottom of page 4, two redundant sentences containing "molecular mechanisms".

-On Page 7, I found the description of the rescue experiment lacking enough detail and hard to follow. Can the authors describe the mis18-262 mutant? This is the first time it is mentioned and there is no reference or description of its phenotypes. Why was this particular mutation chosen? Also, the legend mentions thiamine, it would help if the text said a little bit more about how the experiment was done (plasmid used for expression, is it overexpressed or expressed at low levels, why the -ura, etc.)

-Figure 1D, it would help the reader easily identify the region impacted by the mutations if the residues mutated were highlighted in 1D, and not just in the alignment in 1B. In 2A too, the amino acids mediating the interface are shown, the ones that were mutated, 131, 22, 114 could be circled for clarity.

-The legend for figure 2 contains many details that are redundant with the main text. Also, I suggest rewording the heading- "innate tendency to form..."- to something like "conserved (or intrinsic) preference (or ability) to form...."

-The authors switch between spMis18MeDiY and spMis181-120 in text and figures, it would be a good idea to be more consistent with the nomenclature for clarity.

-I found it confusing to call Mis18fl (which in my mind recalls a wild type protein) something that has mutations. I suggest calling the full-length proteins containing mutations with a different name such as Mis18FL-131A, etc.

-I131A and Y114A mutants don't seem to be enriched at cc2 by ChIP, however by immunofluorescence they look indistinguishable from wt Mis18-GFP (Figure S4), why? At what cell cycle stage are the cells depicted? Can the authors provide an interpretation of this result?

-On page 11 the wording is unnecessarily confusing: "Dimer disrupting mutations I31A and Y114A abolished the ability of Mis18fl to rescue growth at 36oC in both mis18-262 and mis18-818 cells". This is the same as saying that I31A and Y114A mutants cannot rescue the viability defect of mis18-262 and mis18-818 at the restrictive temperature.

-Similarly the sentence "as the Mis18MeDiY I31A and Y114A mutants failed to negatively influence growth at semi-permissive temperature." I suggest using an active tense here such as "Mis18MeDiY I31A and Y114A mutants did not have a negative effect on growth"

Referee #2:

In most eukaryotes, centromere identity is defined by the presence of a histone H3 variant, CENP-A. The epigenetic propagation of the centromere requires the targeted deposition of new CENP-A molecules, which depends on the Mis18 complex and the HJURP/Scm3 CENP-A-specific chaperone. Despite the prior discovery of these molecules and their implication in CENP-A deposition, there is relatively little mechanistic, biochemical, and structural information for how these proteins act. For this paper, the authors have solved the structure of a critical region of fission yeast Mis18 and demonstrate that this region forms a dimer (and subsequently allows formation of a tetramer when present in full length Mis18). The authors conduct a beautiful combination of structural biology, biochemistry to test the oligomerization state of this region and the behavior of mutants, and complementary yeast genetics to test the consequences of selected mutants in vivo. In addition, they conducted limited tests on the human Mis18 proteins to demonstrate that they likely have related structures and properties in this region.

The combined data in this paper is strong and clear, and it provides important information for considering the structure and properties of this critical complex. As a reviewer, I feel the obligation to find the holes in a paper, or suggest experiments that would improve the overall advance or the impact of a paper. However, in this case, I don't have experiments or changes that I feel are necessary. I enjoyed reading this paper, I found the data interesting and useful, and I would

congratulate the authors on the excellent work. I find this paper suitable for publication in EMBO Reports.

Referee #3:

Centromeres are specified by sequence-independent epigenetic mechanisms and CENP-A is a key epigenetic marker for the centromere specification. Fission yeast Mis18 is required for deposition of CENP-A into centromeres. Although functional role of Mis18 is clear, molecular mechanisms how Mis18 is involved in the CENP-A deposition is still unclear. To gain insight for mechanisms of the centromere specification via CENP-A, Subramanian et al. characterized fission yeast Mis18 in this paper. They determined crystal structure of the N-terminal Yippee-like domain of *S. pombe* Mis18 and showed the Yippee-like domain forms a dimer. Mutation of the dimer interface is crucial for centromere localization and function of Mis18 in *S. pombe*. In addition, they demonstrated that the C-terminal domain of Mis18 is involved in tetramer formation of Mis18. They also used human homologues of Mis18 in some analyses and proposed that character of Mis18 is conserved.

This is a solid structural and biochemical work and will contribute to understanding of mechanisms for the centromere specification. However, before publication, they should address some concerns.

1. Although analysis of the Yippee-like domain is clear, functional role of C-terminal domain was a bit unclear. Does spMis18c-term- α make a tetramer by it own? Please clarify this.
2. Is it possible to identify critical sites for tetramer formation in spMis18c-term- α ? If they identify these sites, mutation studies for these sites would be helpful to understand the role of the C-terminus of Mis18.
3. On p10 last line, were Mis18meDIY and Mis18c-term- α eluted at 15.3ml and 12.3ml, respectively? (Figure 3)
4. Although they said that mutations of dimer interface (I31A, Y114A) caused mislocalization of Mis18-GFP (FigureS4), localization data of Figure S4 was not clear. They also conclude this point based on ChIP experiments (Figure 4D). It may be better to show a typical gel image in addition to the graph.
5. I am curious about CENP-A localization deposition in cells expressing I31A or Y114A mutants. Is it possible to examine this?

1st Revision - authors' response

19 January 2016

The point-by-point response to reviewer's comments is as follows:

Referee #1:

Mis18 and the Mis18 complex are key regulators of centromere specification from yeast to vertebrates. However, little is known about the precise function of this protein family.

Through a combination of structural biology and yeast genetics, Subramanian et al. identify a conserved portion of the Mis18 Yippee-like domain that mediates dimerization and oligomerization in vitro and in vivo in *S. pombe*. Additionally, they show an essential role for a 'binding pocket' region present in the Yippee domain. The authors show that human Mis18a, and Mis18b also form hetero-dimers, demonstrating conservation of the role of the Yippee-like domain in mediating dimerization. The specific residues within the Yippee-like domain mediating the dimerization are identified. Mutations in these residues fail to rescue lethality of temperature-sensitive Mis18 mutants. Additionally, dimerization defective Mis18 mutants are unable to bind centromeric chromatin.

The authors did not identify the target of the Mis18 substrate-binding pocket, but this is understandably challenging and beyond the scope of the present work.

The minor comments below focus primarily on improving the clarity of the paper and should be easily addressed during revision.

-The abstract opens with a sentence that is only relevant to humans. I suggest specifying that or re-wording to be either more general or specific to pombe, which is the main system relevant here.

We agree with the reviewer and have made an appropriate change. The revised abstract opens as follows: “Mis18 is a key regulator responsible for the centromere localization of the CENP-A chaperone Scm3 in *S.pombe* and HJURP in humans, which establishes CENP-A chromatin that defines centromeres”.

-Introduction, page 4 near the top: "Exclusion of the Mis18 complex and HJURPScm3 from centromeres during mitosis provides an opportunity for the CENP-ACnp1 loading cycle to reset"; what are the authors trying to say?

We have now altered this sentence to state: “Exclusion of the Mis18 complex and HJURP^{Scm3} from centromeres during mitosis likely provides an opportunity for the CENP-A^{Cnp1} loading cycle to reset and thereby prevent continual CENP-A^{Cnp1} deposition [16].”

-Same page: "As in S. pombe, the human Mis18 complex is required for HJURP recruitment to centromeres, where it deposits...", the wording here suggests Mis18 deposits CENP-A rather than HJURP.

This sentence has now been rephrased to “As in *S. pombe*, the human Mis18 complex is required for HJURP recruitment to centromeres, where CENP-A is deposited during early G1 rather than G2 [15]”

-Bottom of page 4, two redundant sentences containing "molecular mechanisms".

The second sentence has been rephrased to “Mis18 is critical for the specification of centromeres from fission yeast to humans, however, what allows Mis18 to regulate centromere specification remains largely unknown.”

-On Page 7, I found the description of the rescue experiment lacking enough detail and hard to follow. Can the authors describe the mis18-262 mutant? This is the first time it is mentioned and there is no reference or description of its phenotypes. Why was this particular mutation chosen? Also, the legend mentions thiamine, it would help if the text said a little bit more about how the experiment was done (plasmid used for expression, is it overexpressed or expressed at low levels, why the -ura, etc.)

More details on the rescue experiment have been added to the main text: “To test if the putative substrate-binding pocket was required for *spMis18* function *in vivo*, we tested the ability of additional *spMis18* expressed from a plasmid to complement the growth phenotype of *mis18-262* (G117D), cells which exhibit loss-of-function for *spMis18* at the restrictive temperature (36°C) [9]. While expression of wild-type *spMis18* restored growth at 36°C, expressing the pocket mutant (Y74A/Y90A/T105A/S107K, Fig EV1D) failed to complement the loss of *spMis18* function, demonstrating the requirement of this pocket for Mis18 function (Fig 1E).”

Details on selection of the pocket mutant plasmids and conditions for Mis18 expression from the *nmt41* promoter are described in the Materials & Methods section, under the subheadings “Plasmids & *S. pombe* strains” & “Genetic complementation assays”. The legends for Figures 1E, 4B-C & EV4B have been modified to further detail the conditions used for expression of the *spMis18* mutants in complementation assays. A strain table listing genotypes of *S. pombe* strains used in the study is also included (Table EV1).

-Figure 1D, it would help the reader easily identify the region impacted by the mutations if the residues mutated were highlighted in 1D, and not just in the alignment in 1B. In 2A too, the amino acids mediating the interface are shown, the ones that were mutated, 131, 22, 114 could be circled for clarity.

In agreement with the reviewer, we have now highlighted the putative substrate-binding pocket and dimer interface residues mutated in this study in Fig 1D and Fig 2A and D, respectively.

-The legend for figure 2 contains many details that are redundant with the main text. Also, I suggest

rewording the heading- "innate tendency to form..."- to something like "conserved (or intrinsic) preference (or ability) to form...."

We agree with the reviewer. We have now removed some of the redundant information from the legend for Fig 2. We have also changed the heading as suggested by this reviewer.

*-The authors switch between *spMis18MeDiY* and *spMis181-120* in text and figures, it would be a good idea to be more consistent with the nomenclature for clarity.*

We define *spMis18₁₋₁₂₀* as *spMis18_{MeDiY}* only in the section titled ‘**Yippee-like globular domains of Mis18 proteins possess an intrinsic ability to form dimers**’ where we demonstrate that Mis18 Yippee-like domains have the intrinsic ability to form dimers. We have now made sure that in the subsequent sections, *spMis18₁₋₁₂₀* is consistently referred to as *spMis18_{MeDiY}*.

*-I found it confusing to call *Mis18fl* (which in my mind recalls a wild type protein) something that has mutations. I suggest calling the full-length proteins containing mutations with a different name such as *Mis18FL-131A*, etc.*

As suggested, full-length mutant proteins are now referred to as *Mis18_nI31A* and *Mis18_nY114A*.

*-I31A and Y114A mutants don't seem to be enriched at *cc2* by ChIP, however by immunofluorescence they look indistinguishable from wt *Mis18-GFP* (Figure S4), why? At what cell cycle stage are the cells depicted? Can the authors provide an interpretation of this result?*

spMis18-GFP when expressed at endogenous levels decorates the clustered centromeres resulting in a single focus in interphase cells (see Hayashi et al. 2004). The purpose of this experiment was to show that GFP-tagged *spMis18_n*, *spMis18_nI31A* and *spMis18_nY114A* localize to the nucleus. Because each of these *spMis18-GFP* proteins is overexpressed, a diffuse nuclear signal rather than a single focus is observed. Nevertheless, ChIP analyses allow us to assess association of these *spMis18-GFP* proteins specifically with centromeres, and reveal that overexpressed wild-type *spMis18_n-GFP* protein associates with centromeres whereas GFP-tagged *spMis18_nI31A* and *spMis18_nY114A* mutant proteins do not. The cells shown in Fig EV4A are in G2.

*-On page 11 the wording is unnecessarily confusing: "Dimer disrupting mutations I31A and Y114A abolished the ability of *Mis18fl* to rescue growth at 36oC in both *mis18-262* and *mis18-818* cells". This is the same as saying that I31A and Y114A mutants cannot rescue the viability defect of *mis18-262* and *mis18-818* at the restrictive temperature.*

As per the reviewer's suggestion, the text has been modified.

*-Similarly the sentence "as the *Mis18MeDiY I31A* and *Y114A* mutants failed to negatively influence growth at semi-permissive temperature." I suggest using an active tense here such as "*Mis18MeDiY I31A* and *Y114A* mutants did not have a negative effect on growth"*

As per the reviewer's suggestion, the text has been modified.

Referee #2:

In most eukaryotes, centromere identity is defined by the presence of a histone H3 variant, CENP-A. The epigenetic propagation of the centromere requires the targeted deposition of new CENP-A molecules, which depends on the Mis18 complex and the HJURP/Scm3 CENP-A-specific chaperone. Despite the prior discovery of these molecules and their implication in CENP-A deposition, there is relatively little mechanistic, biochemical, and structural information for how these proteins act. For this paper, the authors have solved the structure of a critical region of fission yeast Mis18 and demonstrate that this region forms a dimer (and subsequently allows formation of a tetramer when present in full length Mis18). The authors conduct a beautiful combination of structural biology, biochemistry to test the oligomerization state of this region and the behavior of mutants, and complementary yeast genetics to test the consequences of selected mutants in vivo. In addition, they conducted limited tests on the human Mis18 proteins to demonstrate that they likely have related structures and properties in this region.

The combined data in this paper is strong and clear, and it provides important information for considering the structure and properties of this critical complex. As a reviewer, I feel the obligation to find the holes in a paper, or suggest experiments that would improve the overall advance or the impact of a paper. However, in this case, I don't have experiments or changes that I feel are necessary. I enjoyed reading this paper, I found the data interesting and useful, and I would congratulate the authors on the excellent work. I find this paper suitable for publication in EMBO Reports.

Referee #3:

Centromeres are specified by sequence-independent epigenetic mechanisms and CENP-A is a key epigenetic marker for the centromere specification. Fission yeast Mis18 is required for deposition of CENP-A into centromeres. Although functional role of Mis18 is clear, molecular mechanisms how Mis18 is involved in the CENP-A deposition is still unclear. To gain insight for mechanisms of the centromere specification via CENP-A, Subramanian et al. characterized fission yeast Mis18 in this paper. They determined crystal structure of the N-terminal Yippee-like domain of *S. pombe* Mis18 and showed the Yippee-like domain forms a dimer. Mutation of the dimer interface is crucial for centromere localization and function of Mis18 in *S. pombe*. In addition, they demonstrated that the C-terminal domain of Mis18 is involved in tetramer formation of Mis18. They also used human homologues of Mis18 in some analyses and proposed that character of Mis18 is conserved.

This is a solid structural and biochemical work and will contribute to understanding of mechanisms for the centromere specification. However, before publication, they should address some concerns.

1. Although analysis of the Yippee-like domain is clear, functional role of C-terminal domain was a bit unclear. Does spMis18c-term- α make a tetramer by it own? Please clarify this.

We thank this reviewer for raising this interesting question. To address this, we have carried out SEC-MALS analysis of His-GFP-spMis18_{c-term- α} (Untagged spMis18_{c-term- α} (8.7 kDa) is not big enough to make reliable mass measurement by SEC-MALS). It is worth noting that the variant of GFP that we have used (EGFP) has been demonstrated elsewhere not to have the ability to oligomerize. The measured molecular weight unambiguously demonstrates that spMis18_{c-term- α} is a trimer. This data is now included as Fig 3D.

2. Is it possible to identify critical sites for tetramer formation in spMis18c-term- α ? If they identify these sites, mutation studies for these sites would be helpful to understand the role of the C-terminus of Mis18.

We agree with the reviewer that it is important to identify the critical sites for tetramer formation in spMis18_{c-term- α} . However, as the editor had correctly pointed out, addressing this question is beyond the scope of the work presented here.

3. On p10 last line, were Mis18meDIY and Mis18c-term- α eluted at 15.3ml and 12.3ml, respectively? (Figure 3)

We thank the reviewer for pointing this out. We have now corrected this in the revised manuscript.

4. Although they said that mutations of dimer interface (I31A, Y114A) caused mislocalization of Mis18-GFP (Figure S4), localization data of Figure S4 was not clear. They also conclude this point based on ChIP experiments (Figure 4D). It may be better to show a typical gel image in addition to the graph.

We have addressed the apparent discrepancy between the immunolocalization and ChIP data in our response to Reviewer 1's comments above, and modified the text to clarify our conclusions from the ChIP and imaging experiments. Based on our ChIP results from Fig 4D, we conclude that the dimer interface mutants spMis18_nI31A and spMis18_nY114A show reduced centromere association, and not that they are mislocalized. Fig EV4A shows that the mutants are still nuclear in nature, much like the wild-type spMis18_n protein: we do not draw any additional conclusions from Fig EV4A.

The data presented in Fig 4D results from quantitative ChIP generated by real-time PCR analysis. The use of quantitative real-time PCR assays for ChIP analyses has been the gold standard for quantification of ChIP experiments for several years (e.g.; Saunders *et al* Science 301(5636):1094-96 (2003); Raisner *et al* Cell 123:233-248 (2005), Joshi & Struhl Mol Cell 20:971-78 (2005), Schalch *et al* Mol Cell 34:36-46 (2009), Moser *et al* EMBO J. 28:810-820 (2009)). The chromatin field has almost completely moved away from the old-style semi-quantitative gel analysis of IP'd DNA. We do not see it as being in anyway beneficial to attempt to verify our reproducible (performed in triplicate), quantitative analyses with suboptimal semi-quantitative gel-based assays.

5. I am curious about CENP-A localization deposition in cells expressing I31A or Y114A mutants. Is it possible to examine this?

We have performed ChIP assays for CENP-A^{Cnp1} in cells expressing the *spMis18_nI31A* & *spMis18_nY114A* mutant proteins. Essentially no alteration in CENP-A^{Cnp1} levels at centromeres is detected. Since the *spMis18_nI31A* & *spMis18_nY114A* mutant proteins are unable to associate with centromeres, it is not surprising that they do not affect CENP-A^{Cnp1} deposition. These data have been added to the manuscript as Fig EV4C and are mentioned in the text.

2nd Editorial Decision

03 February 2016

I am very pleased to accept your manuscript for publication in the next available issue of EMBO reports. Thank you for your contribution to our journal.

REFEREE REPORTS

Referee #3

Authors have done several critical experiments to address various concerns from reviewers. I am satisfied with their responses to all comments from reviewers. As the paper contains many important points for centromere-specification, I recommend the paper for publication in EMBO R.

Corresponding Author Name: A. Arockia Jeyaprakash

Manuscript Number: EMBOR-2015-41520V1